# Don't dumb it down: The effects of jargon in COVID-19 crisis communication

**Hillary C. Shulman**[ID]*, **Olivia M. Bullock**

School of Communication, The Ohio State University, Columbus, OH, United States of America

* shulman.36@osu.edu

## Abstract

Experts are typically advised to avoid jargon when communicating with the general public, but previous research has not established whether avoiding jargon is necessary in a crisis. Using the ongoing COVID-19 pandemic as a backdrop, this online survey experiment ($N =$ 393) examined the effect of jargon use across three different topics that varied in situational urgency: COVID-19 (high urgency), flood risk (low urgency), and federal emergency policy (control). Results revealed that although the use of jargon led to more difficult processing and reduced persuasion for the two less-urgent topics (flood risk, emergency policy), there was no effect of jargon in the COVID-19 condition. Theoretically, these findings suggest that the motivation to process information is an important moderator for crisis communication in particular and science communication in general. Practically, these findings suggest that science communicators, during times of crisis, do not need to "dumb down" their language in the same way they should during non-crises.

**Data Availability Statement:** Data are available in OSF: (https://osf.io/gb5nc/).

**Funding:** this research was funded through the lead author's personal research funds, granted through the School of Communication at Ohio

## Introduction

A popular refrain in science communication is to avoid scientific language, otherwise known as jargon, when communicating with the general public [1–3]. Indeed, an abundance of research has revealed that the presence of jargon is alienating [4], undermines comprehension [5], reduces engagement [6, 7], and creates a barrier to entry in certain fields (e.g., STEM) [6, 7]. Although the negative effects of jargon are well-documented, jargon serves important functions as well. Jargon, by definition, conveys information in the most precise and efficient way possible [8]. And, in circumstances where precision and timing are paramount, such as during a crisis, communicating in the most accurate way can mitigate costly miscommunications. Here, rather than take an all-or-nothing approach to the utility of jargon in public-facing science communication, guided by feelings-as-information theory [9] and the elaboration likelihood model [10], we seek to address whether the convention to avoid jargon in science communication generalizes to crisis communication as well.

COVID-19 offers a particularly important ecological context to test the persuasive impact of jargon use in science versus crisis communication. In the early days of the COVID-19 pandemic in the United States, news was rapidly circulating about how the virus spread, who was vulnerable, and what happened to those who became infected. Beyond health information, policy-related discussions soon entered the public discourse, including how to control this

State University. That said, it remains that the funders had no role in the study design, data collection and analysis, decision to publish, or the preparation of the manuscript. Moreover, neither author received a salary for this work

**Competing interests:** The authors have declared that no competing interests exist.

dangerous disease and the economic impact of these decisions. Together, information about COVID-19 quickly became complicated, scientific, political, economic, and–most important to the present study–unfamiliar. At face value, information about COVID-19 includes all the complicated features of science communication. The key difference, however, is that given the threat and urgency of the pandemic, people should be more motivated to learn about COVID-19 relative to other, less pressing, issues. Thus, the current context offers a timely theoretical and practical opportunity to tease apart whether the negative effects of jargon can be attributed to jargon itself, or to the public's lack of motivation to process complex information under ordinary circumstances. By improving our understanding of *why* and *when* jargon should be used, scholars and practitioners can design more persuasive and effective messages.

## Background

### Explaining the negative effects of jargon

Jargon can be defined as "the technical terminology or characteristic idiom of a special activity or group" [1, 8]. Because jargon is specific to a group and highly technical in nature, jargon is typically considered a counterproductive way to deliver information to general audiences. Indeed, prior research suggests that jargon terms undermine peoples' ability to accurately understand message content [4, 5, 11] and make it feel more difficult to process the information presented [4, 11]. Because people tend to dislike effortful processing [12, 13], when information looks complicated, individuals are prone to ignoring or discrediting this information rather than meaningfully engaging with the material [6, 12–15]. In the context of science communication, recent experimental work [4, 11] revealed that the presence of jargon functions as a cue that signals that the presented information will be effortful to process. When participants saw this cue, they became more likely to disregard and resist the information provided, even when definitions explaining the jargon were presented [4, 11]. All told, these results reveal that for topics outside one's expertise, a difficult processing experience will reduce an audience's engagement with persuasive messaging. To explain the theoretical framework underlying this phenomenon, work in metacognition is reviewed below.

Research in metacognition offers that peoples' feelings while processing information (e.g., their emotional state, confidence, or experience of ease versus difficulty) affect people's judgments towards the information itself [9, 14, 16, 17]. This is the first proposition of feelings-as-information theory (FIT) [9], which is one theoretical lens guiding this experiment. In particular, this study focuses on one form of metacognition, called processing fluency, which refers to the effort that accompanies one's information processing experience [9, 16, 17], or, in other words, how hard someone feels they have to work in order to understand a message. Processing fluency experiences can range from difficult and effortful (disfluent) to easy (fluent). Prior research guided by FIT [4, 6, 11, 12] has found that jargon-laden messages lead to a more disfluent experience than messages that do not contain jargon. And, as a result of these feelings of effort, people reported less interest, liking, and engagement with the scientific topics under investigation. The association between effortful processing and negative emotions exists because the negative feelings associated with the need to exert unintended or unwanted effort become attached to the information being presented [9, 13–17]. And, as a result of this attachment, feelings of effort become associated with disinterest, disliking, and decreased engagement with the subject matter [9, 13]. This is a reliable finding in the metacognition literature [13–17] and offers theoretical support for the convention in science communication to keep information simple when communicating with general audiences.

The contribution of this work is testing whether the aforementioned theoretical proposition and conventional rules of thumb about avoiding jargon translate to crisis communication.

Notably, work from science communication [4, 11], as well as health [15] and political communication [12, 18], tend to assume that audiences are unmotivated to expend the energy necessary to process complicated information outside of their areas of expertise or interest. As such, a lack of motivation, coupled with effortful processing, promotes negativity towards the subject matter. However, as we argue below, in a crisis, the assumption that people do not want to expend energy on processing crisis-related information may be misguided. Thus, disentangling the effects of jargon from the effects of motivation to process information becomes critical to understanding both why and when jargon should, or should not, be used. We explore these ideas below.

## The moderating role of motivation

Guided by the elaboration likelihood model (ELM) [10], the motivation to process information should moderate the effect of jargon and processing fluency on persuasive outcomes. According to the ELM, people process new information in one of two styles: The systematic and deliberate style known as central processing, or the quick and heuristic style known as peripheral processing. When people process information centrally, they thoughtfully consider the information presented and are willing to put forth the effort into scrutinizing its quality. Conversely, when people process peripherally, they do not thoughtfully consider the informational substance and instead rely on cues, or heuristics, to arrive at a judgment. A key determinant of whether people process information centrally or peripherally is motivation: When people are motivated to process information carefully, they focus on message substance, and when people are not motivated, they focus on message style. Relating this proposition to the current context, when people are motivated to process information, the presence or absence of jargon and the resulting information processing experience should be meaningless because people should be focused on the message substance (i.e., the content of the message) rather its style (here, whether it contains jargon). On the other hand, when motivation to process is low, then style features such as the presence or absence of jargon should impact one's processing fluency and subsequent judgments towards the topic under investigation. This expectation is supported both by the ELM [10] and FIT [9].

To test the boundary condition of motivation to process information, this experiment examines the effect of jargon on processing fluency and persuasive outcomes across three different topics that vary in situational urgency–an ecological proxy for motivation. Each message topic aimed to increase audiences' awareness of some future risk, and to offer behavioral guidelines to help mitigate these risks. To assess message (in)effectiveness, four outcomes critical to persuasion were measured: Motivated resistance to persuasion, which measured participants' motivation to counterargue and exhibit reactance against the message [19]; credibility, which measured perceptions of message accuracy and trustworthiness [20]; perceived risk, which assessed perceptions of how negatively the threat would impact society [21]; and perceived severity, which assessed perceptions of how negatively the threat would impact the individual [22]. Taken together, the messages created here would be considered more effective, or persuasive, if word choice and accompanying processing fluency served to reduce motivated resistance to persuasion, and increased perceptions of credibility, risk, and severity. That said, recall that the presence of jargon tends to negatively impact persuasion, so, as our hypothesis articulates below, we expect that–for the non-urgent topics–the presence of jargon will undermine, not facilitate, persuasive efforts.

Thus, our primary study hypothesis states that as motivation to process information increases (via control, low urgency, high urgency conditions, respectively), the indirect effect of jargon through processing fluency on outcomes will weaken (H1).

## Materials and method

### Sample

This survey experiment ($N$ = 393) was hosted by CloudResearch [23] from March 24 until April 4, 2020 ($M_{\text{Age}}$ = 40.26, $SD$ = 18.46, 44.5% Female). During this time, most states in the U. S. were in the early stages of lockdown to reduce the spread of COVID-19; therefore, the dissemination of virus-related information was still rapidly evolving. To best ensure data quality [24], MTurk workers qualified for participation if they were located within the United States and had obtained at least a 95% completion rate on at least 500 HITS or were designated as Masters status; could pass a CAPTCHA and an attention check; and could respond to an open-ended question. There were no respondents with complete data who failed these tests, and therefore all data were included in analyses. On average the survey took 20.40 minutes to complete. Participants were compensated either $0.80 ($n$ = 160) or $2.00 ($n$ = 233) for their participation. These reward values differed for practical reasons. When HITS were not being completed in a timely fashion with the lower reward, it was decided to increase this rate to incentivize timely participation, given that situational urgency was a key factor in this investigation. Because two different samples were used to facilitate faster data collection, this sample information was included as a covariate in hypothesis testing. All study materials and procedures were approved and determined exempt by the Institutional Review Board (IRB) at The Ohio State University's Office of Responsible Research Practices (#2020E0274). Before the study began, participants were first presented with an online consent form approved by the IRB that stated that by clicking on the "proceed" arrow, participants were providing their informed consent to participate.

### Experimental design

To examine whether contextual urgency produced the motivation to reason through complex information, we compare the effects of messages containing jargon ($n$ = 197) versus no jargon ($n$ = 196) across three topic conditions that vary in situational urgency: COVID-19 (high urgency, $n$ = 134), flood risk (low urgency, $n$ = 131), and policy information about how the United States handles national emergencies (control, $n$ = 128). Thus, participants were randomly assigned to 1 of 6 experimental conditions using the randomizer function in Qualtrics (S1 Table). All of the information presented was factual and was obtained from credible sources, including organizations such as the Centers for Disease Control and Prevention, the American Academy of Pediatrics, and the Federal Emergency Management Agency, as well as mass media outlets such as the *New York Times*, *POLITICO*, and *WIRED*. Notably, all the jargon words used in this experiment appeared in these materials.

Following the informed consent page, participants were exposed to an introductory paragraph that contextualized the issue, provided an importance statement, and introduced key terms that would be used in the subsequent paragraphs. This statement was identical across the jargon/no-jargon conditions but differed between topics (78 words each). This paragraph was held on-screen for a minimum of five seconds. After five seconds, a "continue" button appeared at the bottom of the screen that allowed participants to advance to the next page when ready. The second page displayed the substance of the information about the topic. In the jargon condition, 20 terms were included that were replaced with simpler terms or explanations in the no-jargon condition, a technique used in previous research [4, 6, 11, 12, 17, 18]. These paragraphs were held on screen for a minimum of eight seconds to better ensure participants read the information (105 words each). The third page included guidelines for how to respond in a crisis. In an effort to make the messages more equivalent across topics, all

participants received the same list of guidelines in either a jargon or no-jargon version depending on language condition assignment. This information was held on-screen for a minimum of eight seconds (98 words each). After exposure to message condition, participants responded to the scales presented below.

## Materials

All scales for this study were taken, or adapted, from published research. Each item used a 1–7 response scale in which higher scores reflect higher agreement with the concept being measured. These scales include processing fluency (six items, $M = 4.96$, $SD = 1.30$, $\alpha = .86$), in which a sample item includes, "the passage felt easy to read" [12]; motivated resistance to persuasion (eight items, $M = 2.87$, $SD = 1.18$, $\alpha = .84$), in which a sample item includes, "I found myself thinking of ways I disagreed with the message I saw" [19]; credibility (four items, $M = 5.87$, $SD = 1.03$, $\alpha = .94$), "The message I saw seemed accurate "[20]; risk perceptions (three items, $M = 5.83$, $SD = 1.10$, $\alpha = .91$), which measured health, safety, and prosperity risk perceptions [21]; and finally, perceived severity (five items, $M = 4.11$, $SD = 1.44$, $\alpha = .87$), in which a sample item includes "my chances of contracting/experiencing [TOPIC] are high" [22]. All scale items can be found in the S1 Appendix, and descriptive statistics by condition are presented in the S2 Table.

## Results

All analyses were conducted using Model 7 in the PROCESS macro for SPSS [25]. The statistical model is illustrated in Fig 1. A separate model was run for each outcome, though the primary independent variable (jargon condition), moderator (topic condition), mediator (processing fluency), and covariate (sample), always remained the same. The data, output, and model building details for these tests are available on the Open Science Framework.

Consistent with expectations, the effect of jargon on processing fluency was moderated by topic (Table 1, $R^2 = .13$). Specifically, the conditional effect of jargon on processing fluency was

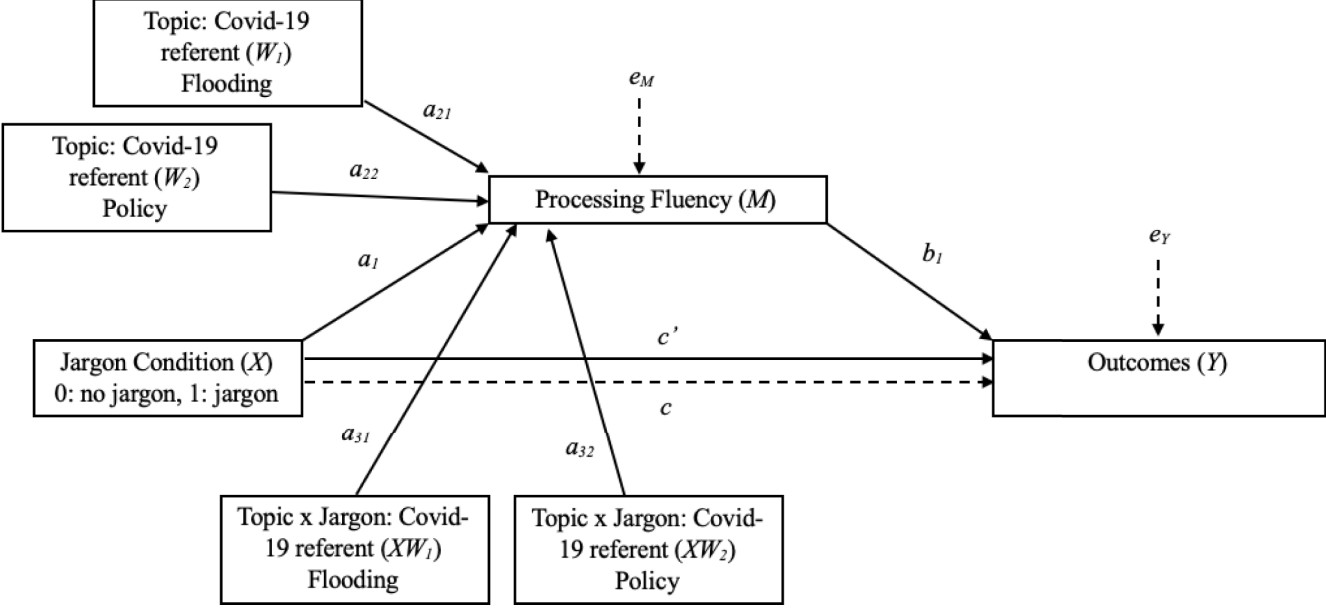

**Fig 1. Hayes' (2013) statistical model diagraming Model 7 in PROCESS.** Estimates for each path can be found in Tables 1 and 2.

**Table 1. Path coefficients between experimental variables and the model mediator.**

| Path Estimates | Processing Fluency B (SE) | 95% Confidence Interval (Lower Limit, Upper Limit) |
|---|---|---|
| Constant (COVID-19 referent) | 5.62 (0.17) | 5.28, 5.97 |
| Jargon ($a_1$) | 0.11 (0.22) | -0.31, 0.54 |
| Flood Topic ($a_{21}$) | -0.29 (0.22) | -0.71, 0.14 |
| Policy Topic ($a_{22}$) | 0.04 (0.22) | -0.39, 0.47 |
| Jargon x Flood ($a_{31}$) | -0.59 (0.31) | -1.19, 0.02 |
| Jargon x Policy ($a_{32}$) | -0.85 (0.31)** | -1.46, -0.24 |
| Sample | -0.67 (0.13)*** | -0.92, -0.41 |
| F | 9.50*** | |
| $R^2$ | .13 | |
| Conditional Effects | | |
| Covid-19 | 0.11 (0.22) | -0.31, 0.54 |
| Flood Risk | -0.47 (0.22)* | -0.90, -0.04 |
| Emergency Policy | -0.74 (0.22)*** | -1.17, -0.30 |

All (*a*) paths estimated with 10,000 Bootstrapped resamples from Hayes' (2013) PROCESS Model 7. These estimates come from the model predicting motivated resistance to persuasion. All models yield slightly different estimates due to Bootstrapping. The sample covariate represents whether participants were compensated $0.80 (0) or $2.00 (1).

\*$p < .05$

\*\*$p < .01$

\*\*\*$p < .001$.

significant, and negative, for the policy topic ($B = -0.74$, $SE = 0.22$, $t = -3.33$, $p < .05$, 95% CI [-1.17, -0.30]), followed by the flood topic ($B = -0.47$, $SE = 0.22$, $t = -2.14$, $p < .05$, 95% CI [-0.92, -0.04]). For the COVID-19 topic, however, the effect of jargon was not significant ($B = 0.11$, $SE = 0.22$, $t = 0.53$, $p = .598$, 95% CI [-0.31, 0.54]). Thus, the first half of the model supported H1; jargon impaired processing fluency, unless the topic was of high urgency. When motivation was high, the presence or absence of jargon did not affect fluency reports.

The second part of H1 proposed that the mediator of processing fluency would significantly, and adversely, affect persuasive outcomes for the policy and flood risk conditions, but not for the COVID-19 condition. All estimates are presented in Table 2 and the indirect effects for the complete model, across topics, are illustrated in Fig 2. In sum, all four mediation models cohered with expectations such that jargon exerted the strongest indirect effect in the control condition, followed by the low urgency condition, while no significant indirect effects were found for the high urgency condition. These results offer experimental support, across four outcomes, for the moderating role of motivation–driven by situational urgency–when people process complex messages outside their area of expertise.

## Discussion

For communicators in the throes of a crisis, it is critical to convey factual, precise information that will also engage the general public. Under normal circumstances, communicating with technical, idiosyncratic words, or jargon, is viewed negatively by audiences [1–9]. In particular, research in science communication and beyond has found that the presence of jargon damages persuasive efforts [9] and can have a disengaging effect on audiences [4–6]. For these reasons and more, science communicators are encouraged to keep it simple. Despite the heuristic appeal of this recommendation, however, jargon is useful when technical information needs to be properly disseminated. Thus, rather than eliminate these terms altogether, this work sought

**Table 2. Path coefficients between experimental variables, the model mediator, and all outcome variables.**

| Path Estimates | Outcomes | | | |
|---|---|---|---|---|
| | Motivated Resistance to Persuasion *B* (*SE*) | Credibility *B* (*SE*) | Perceived Risk *B* (*SE*) | Perceived Severity *B* (*SE*) |
| Constant | 5.61 (0.22)*** | 3.97 (0.24)*** | 4.76 (0.27)*** | 5.87 (0.33)*** |
| Jargon (*c'*) | -0.29 (0.09)** | 0.35 (0.10)*** | 0.33 (0.11)** | 0.01 (0.14) |
| P. Fluency (*b₁*) | -0.56 (0.04)*** | 0.33 (0.04)*** | 0.19 (0.04)*** | -0.39 (0.06)*** |
| Sample | 0.26 (0.10)** | 0.16 (0.10) | -0.03 (0.12) | 0.28 (0.15) |
| *F* | 90.18*** | 25.30*** | 8.50*** | 21.29*** |
| *R²* | .42 | .17 | .06 | .15 |
| Conditional Indirect Effects (*c*) [95% CI] | | | | |
| COVID-19 | -0.06 (0.12) [-0.31, 0.17] | 0.05 (0.07) [-0.10, 0.19] | 0.02 (0.04) [-0.05, 0.11] | -0.05 (0.09) [-0.23, 0.12] |
| Flood Risk | 0.26 (0.13) [0.01, 0.52] | -0.16 (.08) [-0.31, -0.01] | -0.08 (0.05) [-0.20, 0.00] | 0.17 (0.09) [-0.01, 0.36] |
| Emergency Policy | 0.41 (0.12) [0.19, 0.64] | -0.22 (0.08) [-0.39, -0.08] | -.14 (0.05) [-0.25, -0.04] | 0.29 (0.10) [0.12, 0.50] |
| Index of Moderated Mediation [95% CI] | | | | |
| COVID-19—Flood | 0.33 (0.18) [-0.02, 0.68] | -0.20 (0.11) [-0.42, 0.00] | -.11 (0.07) [-.26, 0.01] | 0.23 (0.13) [-0.03, 0.48] |
| COVID-19—Policy | 0.48 (0.17) [0.14, 0.81] | -0.27 (0.11) [-0.49, -0.07] | -0.16 (.07) [-.31, -.04] | 0.35 (0.13) [0.11, 0.62] |

Paths estimated with 95% bias-corrected bootstrap confidence intervals based on 10,000 resamples from Hayes' (2013) PROCESS Model 7. The path estimate that predicts each outcome from jargon is also referred to as the direct effect estimate (c') of jargon on outcomes. For this categorical variable, the no-jargon condition is the referent category. The sample covariate represents whether participants were compensated $0.80 (0) or $2.00 (1).

*$p < .05$

**$p < .01$

***$p < .001$.

to understand if there was a theoretical precedent for when more complicated forms of communication might not undermine strategic efforts. Guided by FIT [15] and the ELM [16], it was argued that motivation to process information might mitigate the negative impact of jargon. The results from this experiment offers evidence that situational urgency–or in other words, a time of crisis–potentially mitigates the impact of jargon use on processing fluency and aversive persuasive outcomes. This finding is consistent with FIT [15] and the ELM [16], and implies that during a time of crisis, experts may not need to "dumb down" information and avoid jargon altogether while speaking to a general audience. During non-crisis times, however, when motivation to process information is likely to be low, results from this study suggest that jargon should still be avoided. Despite support for these ideas, the section below outlines study limitations.

## Limitations

In spite of the theoretical and practical importance of these findings, limitations concerning whether these findings would generalize to other messages, populations, and circumstances merit acknowledgement. For one, the effects we obtained were contingent on only one set of messages that differed by topic, although the third and final aspect of each of our stimulus messages (i.e., the recommended guidelines to follow) was the same across topic conditions. This leaves open the possibility that other message features contributed to the effects observed. For instance, it is possible that given the widespread media coverage of COVID-19, the jargon terms used in the COVID-19 condition were more familiar than the terms used in the flooding and policy conditions. Other possibilities include variability between message topics' level of abstraction, personalization, and salience. Taken together, the fact that these messages varied in ways beyond topic urgency and jargon underscores the need to test these relationships

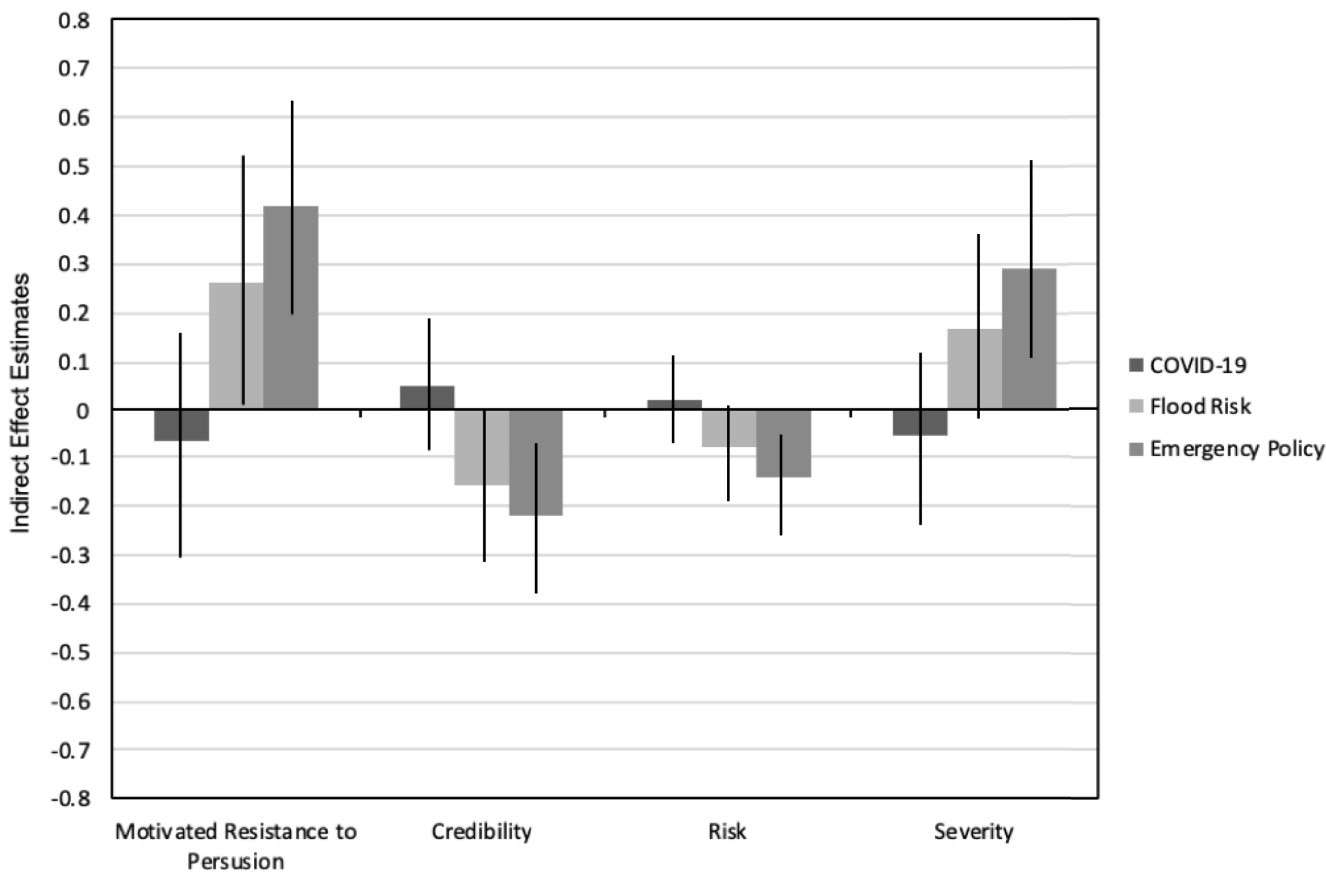

**Fig 2. Conditional indirect effect estimates including 95% confidence intervals by topic.**

across multiple messages, a technique called the message sampling approach [26]. This technique allows for more informed and generalizable conclusions regarding the link between message features and message effects, and should be prioritized in future research.

A second set of limitations pertain to our sample. This data was collected from MTurk rather than a random sample from the general population. Because issues have arisen with MTurk workers regarding data quality, trustworthiness, and replicability [24], whether these findings would generalize to other populations remains an open question. Moreover, there were limitations inadvertently caused by our provided incentive. Given that the size of the incentive was changed from \$0.80 to \$2.00 to expedite data collection, an unintended source of variance was introduced into our models. Specifically, participants receiving more money, and who also took the survey up to one week later, reported a significantly more difficult processing experience ($t$ [380] = 5.23, $p < .001$) than those receiving less money and who took the survey the week before. This trend was even true in the COVID-19 condition ($t$ [127] = 4.45, $p < .001$), which was surprising because one would expect that processing COVID-related information would become easier over time, not more difficult, given the immense media coverage of the issue. Thus, it remains unclear why these differences emerged. It could be that the size of the incentive produced differences in participant motivation, or that the social context changed over the course of a week in ways that affected scores. Both of these possibilities are practically and theoretically interesting, but unfortunately cannot be disentangled here. In sum, given the presence of these methodological and contextual limitations, replicating these

effects using a different set of messages, a different sample, and more consistent methods is critical.

## Conclusion

This investigation sought to understand how best practices in science communication would translate to crisis communication. The results from this experiment suggest that when more urgent, risky, or pressing concerns are communicated, the negative effects of jargon are reduced relative to less urgent topics. Taken together, this study provides ecological evidence, and a practical application, for a well-known assertion in the psychological sciences: When people are motivated to process information, they will.

## Supporting information

**S1 Table. Messages across jargon condition and topic condition.**
(DOCX)

**S2 Table. Descriptive statistics across conditions for all study variables.**
(DOCX)

**S1 Appendix. Items for all scales used in analyses.** An asterisk next to an item indicates the item was removed to improve reliability.
(DOCX)

## Acknowledgments

The corresponding author would like to thank her partner, David DeAndrea, for wrangling two very small children while this study (and this writing) was in progress. Those precious moments every day made this possible.

## Author Contributions

**Conceptualization:** Hillary C. Shulman, Olivia M. Bullock.

**Data curation:** Hillary C. Shulman.

**Formal analysis:** Hillary C. Shulman.

**Methodology:** Hillary C. Shulman, Olivia M. Bullock.

**Writing – original draft:** Hillary C. Shulman.

**Writing – review & editing:** Olivia M. Bullock.

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
