## [Decision Letter · Decision Letter 0]

23 Jul 2020

PONE-D-20-18585

Don't dumb it down: The effects of jargon in COVID-19 crisis communication

PLOS ONE

Dear Dr. Shulman,

Thank you for submitting your manuscript to PLOS ONE.  Although the research reported in this manuscript is important and interesting and has merit, we feel that it does not fully meet PLOS ONE’s publication criteria as it currently stands. Therefore, we invite you to submit a revised version of the manuscript that addresses the points raised during the review process.The primary concerns are summarized below;  additional issues can be seen in the reviewers’ comments. A revision would need to address all of these (or provide a justification for not addressing).

Major issues:

First,  as noted by Reviewers 1 and 3, there are confounds for your urgency variable in that the urgent message differs from the less urgent messages in multiple ways (familiarity, salience, personalization, etc.).  This is a serious limitation that should be noted, and your conclusions tempered accordingly.

Second, power is not mentioned until the Discussion when  it’s noted that the study involved a smaller-than-intended sample size. The power and/or sample size considerations should be noted in the Method section and the criteria for describing the study as under-powered should be made explicit.

Third, the sample consists of  MTurk workers and so there should be some mention of the limitations (e.g., non-naivety and trustworthiness) of participants from this platform.

Fourth, the analyses are presented in a clear and straightforward manner.  However, readers would likely appreciate more descriptive data (means and standard deviations) regarding your critical variables (e.g., fluency as a function of jargon and urgency; but see also Reviewer 1’s request for means and standard deviations for your outcome variables); if not presented in the text they could be included in your supplementary files.

We look forward to receiving your revised manuscript.

Kind regards,

Thomas Holtgraves, Ph.D.

Academic Editor

PLOS ONE

Journal Requirements:

2. Please provide additional details regarding participant consent. In the Methods section, please ensure that you have specified (1) whether consent was informed and (2) what type you obtained (for instance, written or verbal). If your study included minors, state whether you obtained consent from parents or guardians. If the need for consent was waived by the ethics committee, please include this information.

"No. The funders had no role in study design, data collection and analysis, decision to publish, or preparation of the manuscript."

Reviewers' comments:

Reviewer's Responses to Questions

**Comments to the Author**

1. Is the manuscript technically sound, and do the data support the conclusions?

Reviewer #1: Yes

Reviewer #2: Partly

Reviewer #3: Partly

2. Has the statistical analysis been performed appropriately and rigorously? 

Reviewer #1: Yes

Reviewer #2: Yes

Reviewer #3: Yes

3. Have the authors made all data underlying the findings in their manuscript fully available?

Reviewer #1: Yes

Reviewer #2: No

Reviewer #3: Yes

4. Is the manuscript presented in an intelligible fashion and written in standard English?

Reviewer #1: Yes

Reviewer #2: Yes

Reviewer #3: Yes

5. Review Comments to the Author

Reviewer #1: Review of PONE-D-20-18585: Don't dumb it down: The effects of jargon in COVID-19 crisis communication.

The manuscript presents a study that examined the effect of jargon on resistance, reactance, credibility, and perceived risk across three different topics associated with differing levels of urgency. Whereas jargon use affected processing of information associated with low urgency, jargon use has no discernible influence for high-urgency information (compared to a high urgency control information).

Overall, the paper is well written and organized. The authors address an interesting and timely topic.

The prediction that increased processing from increased motivation or ability has been borne out in many studies. In most cases, these predictions focus on differences in an outcome (e.g., resistance, attitude change, etc.). With jargon having no difference in resistance and credibility in the high urgency message, it implies that, from a practical standpoint, jargon is not needed for a high urgency message to be effective. Is this the case?

Another question the authors may want to address relates to the urgency and jargon manipulations. Specifically, the issue is different across levels of the manipulation. This is reasonable, but one can imagine that the difference in topic also affects other variables (e.g., experience with topic, salience via news exposure, etc.). Importantly, the jargon would also differ across the topics. Perhaps the authors could speak to that.

Another question I have relates to the conceptualization of counterarguing. The term (which refers to actively refuting claims made in the message) seems to be used a catch-all phrase for thought-based resistance. But, in reality, there are different forms of resistance beyond counterarging (e.g., thought bolstering, source derogation, etc.). That said, when describing the research on Page 4, line 70 (references 4 & 11), is it the case that participants counterargued. In other words, did they really counterargue under low processing? From reading the description of the materials on page, the general term of resist would be more consistent with what is being measured and what participants did.

The authors are predicting interactions on the relevant outcome variables. If space allows, it would be nice to see those results and subsequent means and standard deviations.

My apologies, but it is unclear how the analyses in the mediational model were conducted with three levels of the topic/urgency variable. Can the authors clarify how this is handled?

Reviewer #2: 1. Is this manuscript technically sound, and do the data support the conclusions?

Yes, to both except: Provide one sentence on how random assignment was accomplished and what was the N per cell.

Also, it is not clear why jargon should influence processing fluency (lines 119, 120). It would be useful to elaborate on this. An individual may work harder in trying to understand an important message but why would this affect fluency?

Also, a 2 X 3 table showing fluency scores would help in the results.

2. Has the statistical analysis been performed appropriately and rigorously?

The results came out nicely. However, for the DV of fluency would it be helpful to determine if the regression coefficients for policy topic (B = -.74) versus the flood topic (B = -.47) were different? If so, it would tie in nicely with the Figure 1 result.

3. Have the authors made all data underlying the finding in the manuscript fully available?

I could not find any data. I found the 6 different scripts used in the experiment. I also found the five ratings scale employed.

4. Is the MS presented in an intelligible fashion and written in standard English?

OK but tighten up the sentences; read MS out loud.

Reviewer #3: The question posed is interesting, the theory and literature considered as background was appropriate, and the article is written in a crisp, clear style. Non-specialists could read it and appreciate the main claims, but would find it more difficult to follow the analyses.

My biggest concern is the conclusion. It seems too strong and should be modified. The independent variable of interest is "urgency" and whether this affects the usual finding that jargon reduces the effectiveness of public policy recommendations. The problem with any singular sampling of topics differing in urgency, especially given that one of those topics was chosen because of its time sensitivity, is that it is always the case that the communications will differ in ways other than the variable of interest (i.e., urgency). COVID is primarily an individual, personal threat, while floods are more regional, less personal, and more of a threat to property. The policy (no urgency) communication is more abstract, as well as less urgent. More people know something about disease dynamics than about hydraulics and geology of flooding. The punchline is that a conclusion claiming that concern about jargon may be less pressing for urgent communications, and might even increase the persuasiveness of the communication, seems too strong a claim to draw from a single set of communications that vary in a number of ways besides urgency.

Another possible threat to internal validity is that respondents had been exposed to a considerable number of expert descriptions of the COVID threat, so that they had some familiarity with the COVID style jargon, in a way that was not true of flooding and disaster policy concerns. None of the concerns I am raising are disqualifying in my opinion, but the author should be more muted in characterizing the conclusions, or indicating limitations of the study.

Minor: (1) Why was the "never happens to me" question removed? How did it affect reliability? (2) Why was there no measure of urgency, or were the Risk questions just that? If so, the differences in urgency are not as strong as would be desirable? (3) It's too bad that the regions of the country where the respondents lived were not known? The COVID threat at the time was centered in the Northeast. Similarly, flood risk varies enormously by where one lives. Questions on prior exposure to flu, floods, other disasters, would have been interesting to include. (4) the different priced samples (80 vs. 200) could be viewed as reflecting a difference in motivation with the 80 cent group the more motivated to attend to communications.

6. PLOS authors have the option to publish the peer review history of their article (what does this mean?). If published, this will include your full peer review and any attached files.

Reviewer #1: No

Reviewer #2: No

Reviewer #3: No

---

## [Author Response · Author response to Decision Letter 0]

17 Aug 2020

Our response to reviewers document has been uploaded to your system.

---

## [Editor Report · Decision Letter 1]

3 Sep 2020

PONE-D-20-18585R1

Don't dumb it down: The effects of jargon in COVID-19 crisis communication

PLOS ONE

Dear Dr. Shulman,

I am writing with regard to your manuscript, "Don't dumb it down: The effects of jargon in COVID-19 crisis communication" (PONE-D-20-18585R1). I have read your revision, and find that you have done a nice job addressing the concerns raised in the first round of reviews. As such, I am taking the action of accepting your manuscript pending some very minor revisions.

There are three small issues that I would like you to address in your revision:

1 -- I think it would be helpful if you could make your data available as excel spreadsheets in addition to the SPSS files. The excel sheets will probably be more accessible to more users than the SPSS files. In addition, please upload your output file as a PDF so that those without SPSS will have access.

2 -- Please proof carefully the data presented in the manuscript. Just spot checking I found one error (the MRTP constant is 5.61 in the output but reported as 5.86 in Table 2).

3 -- In studies like this I believe it is useful to provide readers with the experimental cover story (i.e., instructions for the participants). These could be provided in your supplementary materials.

Once you've addressed these final points, I will take a quick look and send you the official acceptance letter.

We look forward to receiving your revised manuscript.

Kind regards,

Thomas Holtgraves, Ph.D.

Academic Editor

PLOS ONE

---

## [Author Response · Author response to Decision Letter 1]

7 Sep 2020

Please see our response document for replies to specific comments.

---

## [Editor Report · Decision Letter 2]

9 Sep 2020

Don't dumb it down: The effects of jargon in COVID-19 crisis communication

PONE-D-20-18585R2

Dear Dr. Shulman,

We’re pleased to inform you that your manuscript has been judged scientifically suitable for publication and will be formally accepted for publication once it meets all outstanding technical requirements.

Kind regards,

Thomas Holtgraves, Ph.D.

Academic Editor

PLOS ONE
---

## [Editor Report · Acceptance letter]

14 Sep 2020

PONE-D-20-18585R2 

Don’t dumb it down: The effects of jargon in COVID-19 crisis communication 

Dear Dr. Shulman:

I'm pleased to inform you that your manuscript has been deemed suitable for publication in PLOS ONE. Congratulations! Your manuscript is now with our production department. 

Kind regards, 

on behalf of

Dr. Thomas Holtgraves 

Academic Editor

PLOS ONE